# Position Statement about Gender-Neutral HPV Vaccination in Korea

**DOI:** 10.3390/vaccines12101110

**Published:** 2024-09-27

**Authors:** Kyung-Jin Min, Yung-Taek Ouh, Sangrak Bae, Yong-Bae Ji, Jae-Kwan Lee, Jae-Weon Kim, Kwang-Jae Cho, Dong-Hun Im

**Affiliations:** 1Department of Obstetrics & Gynecology, Korea University Ansan Hospital, Ansan 15354, Republic of Korea; mikji78@gmail.com (K.-J.M.); oytjjang@gmail.com (Y.-T.O.); 2Department of Urology, The Catholic University of Korea Uijeongbu St. Mary’s Hospital, Uijeongbu 11765, Republic of Korea; robinbae97@gmail.com; 3Department of Otolaryngology, Hanyang University Medical Center Guri Hospital, Guri 11923, Republic of Korea; yb20000@hanmail.net; 4Department of Obstetrics & Gynecology, Korea University Guro Hospital, Seoul 08308, Republic of Korea; jklee38@korea.ac.kr; 5Department of Obstetrics & Gynecology, Seoul National University Hospital, Seoul 03080, Republic of Korea; kjwksh@gmail.com; 6Department of Otorhinolaryngology, The Catholic University of Korea Uijeongbu St. Mary’s Hospital, Uijeongbu 11765, Republic of Korea; entckj@naver.com; 7Department of Urology, Chosun University Hospital, Gwangju 61453, Republic of Korea

**Keywords:** human papillomavirus, head and neck cancer, oropharyngeal carcinoma, cervical cancer, penile cancer, genital warts, gender-neutral vaccination

## Abstract

Given the rising incidence of human papillomavirus (HPV)-related diseases, including cervical, penile, and oropharyngeal cancers, particularly among men, the implementation of comprehensive HPV vaccination strategies is necessary in South Korea. This position statement advocates the introduction of gender-neutral vaccination (GNV) in the country. It recommends the administration of the HPV vaccine to both men and women aged 9–26 years to prevent a broad spectrum of HPV-related conditions. Specifically, individuals aged 9–14 years are advised to receive two doses of the vaccine, whereas those aged 15–26 years are advised to receive three doses. The optimal age for vaccination is identified as 11–12 years old. Additionally, this statement recommends that women aged 27 years and older be vaccinated based on the discretion of healthcare providers. The introduction of GNV is essential to curb the spread of HPV and reduce the overall burden of HPV-related cancers, making it a critical public health initiative in Korea.

## 1. Introduction

Human papillomavirus (HPV) is one of the most common sexually transmitted infections worldwide, affecting both men and women [1]. Over 200 types of HPV have been identified, approximately 40 of which are transmitted through sexual contact, infecting the anogenital and oropharyngeal regions. Although many HPV infections are transient and asymptomatic, persistent infections with high-risk HPV types can lead to the development of various cancers, with cervical cancer being the most well-known HPV-related malignancy [2]. Additionally, HPV has been implicated in a significant proportion of other cancers, such as penile and vulvar cancers, as well as those that affect both the male and female sexes, particularly oropharyngeal and anal cancers.

Cervical cancer has long been recognized as a major public health concern, prompting extensive screening programs and the development of prophylactic HPV vaccines [3]. The introduction of these vaccines marked a milestone in the prevention of cervical cancer, with studies demonstrating a substantial reduction in the incidence of high-grade cervical lesions and cervical cancer in vaccinated populations. Initially targeted at young women, these vaccines were designed to protect against the most oncogenic HPV types, primarily HPV 16 and 18, which are responsible for most cervical cancer cases [4].

However, emerging evidence has revealed that HPV-related diseases extend beyond cervical cancer. In men, HPV is the etiological agent in most oropharyngeal cancers and in a significant proportion of anal and penile cancers [5]. The incidence of oropharyngeal cancer has markedly increased, particularly in high-income countries, where it is now the most common HPV-related cancer among men [6]. Despite this, there are no routine screening programs for oropharyngeal, anal, or penile cancers in men that are comparable to the Pap smear for cervical cancer in women, leading to late diagnoses and poorer outcomes.

The lack of organized screening for non-cervical HPV-related cancers in men underscores the need for a broader approach to the prevention of HPV infection. Gender-neutral vaccination (GNV) programs, which include both the male and female genders, have been proposed as strategies to reduce the overall burden of HPV-related cancers [7]. By vaccinating both the male and female sexes, these programs aim to reduce HPV transmission within the population, thereby decreasing the incidence of HPV-related cancers in both men and women [8]. GNV likewise helps protect men at risk of developing HPV-related cancers and diseases, particularly men who have sex with men (MSM), who are disproportionately affected by HPV-related anal cancers.

## 2. Burden of HPV-Related Cancers and Diseases

Globally, approximately 730,000 cases of HPV-related cancers are diagnosed annually in both men and women. HPV 16 and 18 are the most prevalent types and are responsible for approximately 72% of all HPV-attributable cancer cases in women and nearly 100% in men [9]. The burden of HPV-related oropharyngeal cancer is particularly high in men, outnumbering women by a significant margin in many countries [1]. In South Korea, the incidence of oropharyngeal cancer has continuously increased, particularly in males [10].

### 2.1. Global Burden of Cervical Cancer

Cervical cancer remains a significant global public health challenge, particularly in low- and middle-income countries (LMICs), where the burden is disproportionately high [11]. In 2020, approximately 604,127 new cases of cervical cancer and 341,831 deaths occurred worldwide [11]. The age-standardized incidence of cervical cancer varies widely, with the highest rates being observed in Eastern and Southern Africa and the lowest in Western Asia and Northern Europe. This disparity reflects a clear socioeconomic gradient such that countries with lower Human Development Index (HDI) values tend to have significantly higher incidence and mortality rates. Specifically, the incidence rates were three times higher and the mortality rates were six times higher in countries with a low HDI than those in countries with a very high HDI. Despite the proven effectiveness of preventive measures, such as HPV vaccination and screening, reduction in cervical cancer incidence and mortality has been slow in LMICs, with some regions even experiencing increasing trends.

### 2.2. Global Burden of HPV-Related and Other Anogenital Cancers

Globally, HPV infection is responsible for approximately 8500 cases of vulvar carcinoma, 12,000 cases of vaginal cancer, 35,000 cases of anal cancer, and 13,000 cases of penile cancer [9]. Much like cervical cancer, the burden of HPV-related anogenital cancers varies significantly by region [9]. Countries with relatively high age-standardized rates (ASRs) of HPV-attributable anogenital cancers (>1.25 per 100,000) are predominantly in Latin America, North America, and Australia, with some being located in Europe and sub-Saharan Africa.

Penile cancer, though relatively rare on a global scale, represents a significant health burden, particularly in developing regions. Annually, approximately 26,000 new cases are diagnosed worldwide, with the highest incidence being observed in areas such as Africa, Asia, and parts of Latin America. The etiology of penile cancer involves two main pathways: one linked to phimosis, chronic inflammation, and lichen sclerosis, and the other associated with HPV infection. HPV-related penile cancers are predominantly caused by high-risk HPV types, with HPV16 being the most prevalent, accounting for about 68.3% of HPV-positive cases [12]. The pooled prevalence of HPV DNA in penile cancers has been estimated at approximately 50.8%, underscoring the substantial role of HPV in the pathogenesis of this malignancy. Given the high burden of disease and the strong association with HPV, the inclusion of males in HPV vaccination programs is imperative to reduce the incidence of penile cancer and other HPV-related diseases.

Anal cancer, with nearly 90% of cases linked to HPV, is distributed almost equally between the male and female sexes globally. In less developed nations, men have a slightly higher incidence, while women are more affected in developed countries [2]. HPV 16 is more predominant in anal cancer than in cervical carcinoma, with HPV 16 and 18 being responsible for 87% of cases and a combined 96% being linked to HPV types 6, 11, 16, 18, 31, 33, 45, 52, and 58. Vulvar and penile carcinomas, though less common, have notable incidence in regions like Europe, Latin America, and India. Vaginal carcinoma is rarer but has a higher HPV-attributable fraction of 78%, with similar HPV type distributions across vulvar, vaginal, and penile cancers.

### 2.3. Global Burden of HPV-Related Head and Neck Cancer

Three key anatomical sites within the head and neck region have been associated with HPV infection: the oropharynx, and to lesser degrees, the oral cavity and larynx. Globally, HPV is responsible for approximately 38,000 cases of head and neck cancer. In contrast to cervical cancer, the incidence of HPV-associated head and neck cancer is markedly higher in developed countries than in less developed regions. Countries with high ASRs of HPV-related head and neck cancer (>1.25 per 100,000) are primarily located in North America and Europe [6]. Approximately 30% of oropharyngeal cancers, predominantly involving the tonsils and base of the tongue, are attributed to HPV infections, accounting for an estimated 29,000 cases annually. The attributable fraction of HPV in oropharyngeal cancers shows significant regional variations, with the highest rates being observed in more developed countries (exceeding 40% in Europe, North America, Australia, New Zealand, Japan, and South Korea), whereas it remains considerably lower (below 20%) and uncertain in many other parts of the world. For HPV-related cancers of the oral cavity (4400 cases annually) and larynx (3800 cases), the data on HPV prevalence are based on limited case series, predominantly from Europe and North America, suggesting an average HPV prevalence of approximately 4% at both sites. In other regions, the HPV-attributable fraction for cancers of the oral cavity and larynx is even lower, ranging from 1 to 2%.

The higher prevalence of HPV 16 in head and neck cancer than in cervical cancer is notable, with HPV 16 and 18 being responsible for 85% of HPV-associated head and neck cancer cases worldwide, while the combined contribution of HPV 6, 11, 16, 18, 31, 33, 45, 52, and 58 accounts for 90% of these cancers.

### 2.4. Global Burden of Anogenital Warts

Anogenital warts, primarily caused by HPV types 6 and 11, represent a significant burden globally, particularly among males. These warts are the most common manifestation of HPV infection, and while they are benign, their impact on quality of life and healthcare systems is substantial. The incidence of anogenital warts varies by region, with developed countries reporting annual rates of 0.1% to 0.2% in the general population [5]. Males, especially those between the ages of 15 and 24, are disproportionately affected, reflecting patterns of sexual activity and the prevalence of HPV in this demographic. The recurrence of warts following treatment is common, adding to the chronic nature of this condition. Without widespread vaccination and public health interventions, anogenital warts among males remain a significant concern, emphasizing the need for vaccination strategies that include both the male and female sexes to curb HPV and related diseases.

### 2.5. Impact of HPV on Sperm Quality and Male Fertility

Sperm quality is a critical determinant of male fertility, and its decline represents a significant public health concern globally. Various factors contribute to impaired sperm quality, including infections, environmental exposures, and lifestyle factors. Among these, human papillomavirus (HPV) infection has emerged as a notable factor influencing sperm quality, particularly sperm motility. Studies have shown that HPV can bind to the head of spermatozoa, leading to reduced progressive motility and, consequently, a decrease in fertility potential. Meta-analyses indicate that infertile men with HPV infection in their semen exhibit significantly lower sperm motility compared to non-infected individuals, highlighting the broader impact of HPV beyond its association with cancers [1].

## 3. HPV-Related Diseases in Korea

A large-scale study conducted in South Korea examined sex differences in the incidence of head and neck cancer [13]. The study’s findings indicate that head and neck cancer is more prevalent and presents a greater risk to men than to women. This disparity highlights the importance of considering HPV GNV in the context of public health, particularly because of male susceptibility to this cancer. These results emphasize the need for further research on the sex-specific factors that influence the incidence and prevention of head and neck cancer, particularly in relation to HPV-related oncogenic pathways.

Although global GNV programs have proven beneficial, South Korea’s specific disease burden must be carefully considered. The incidence of non-cervical HPV-related cancers, such as oropharyngeal cancer, has been increasing among Korean men, with rates rising from 2.7 cases per 100,000 people in 2013 to 3.1 per 100,000 in 2016 [14]. Despite these figures, the HPV vaccination rates for boys remain critically low (0.7–1.3%) compared to the significantly higher coverage among girls [14]. The disparity reflects a gendered perception of HPV, perpetuating the idea that HPV primarily affects women, and contributing to a lack of awareness and urgency around male vaccination.

The prevalence and distribution of HPV genotypes among Korean men with suspected sexually transmitted infections were analyzed, revealing an overall HPV positivity rate of 62.9% [15]. The most common HPV genotypes identified in men were types 6 and 11, which, although classified as low risk, are associated with conditions such as genital warts. Notably, the prevalence of the high-risk genotype 16 increased with age, indicating its potential role in oncogenesis in men. Additionally, a significant proportion of men were infected with multiple HPV genotypes, with some individuals carrying up to 10 different genotypes. In Korea, head and neck squamous cell carcinoma, particularly oropharyngeal carcinoma, was shown to have a significant association with high-risk HPV. Among these, HPV 16 is the predominant genotype, accounting for 84.3% of high-risk HPV-positive cases [16]. The high prevalence of HPV 16 is especially notable in the oropharyngeal region, including the tonsils, where the virus plays a critical role in oncogenesis. Given the prevalence of HPV 16 and its strong link to oropharyngeal cancers, particularly in men, the introduction and promotion of comprehensive HPV vaccination programs, including GNV, are crucial for reducing the burden of these cancers and improving public health outcomes.

Despite the growing recognition of HPV’s impact on both men and women, significant cultural and financial barriers impede the expansion of GNV in South Korea. A qualitative study by Choi et al. revealed that many Korean mothers view HPV vaccination as primarily a preventive measure for cervical cancer, which leads to a lack of awareness and urgency regarding vaccinating boys. The vaccine is often referred to as the “cervical cancer vaccine”, reinforcing the misconception that it is irrelevant for males [17]. Sociocultural factors also play a significant role. There is a general reluctance to discuss sexual health within families, especially between mothers and sons, which contributes to low awareness about HPV transmission and the importance of vaccination for boys. Mothers in the study expressed discomfort in broaching the topic of sexually transmitted infections (STIs) with their sons, a common sentiment in conservative Asian cultures [17].

Expanding the national immunization program to include boys presents several logistical challenges, particularly given the potential doubling of the vaccine-eligible population (Table 1). The cultural and public health hurdles to this expansion must be carefully managed. Barriers to male HPV vaccination often include a lack of knowledge, vaccine hesitancy, and inadequate healthcare provider recommendations [18]. These issues are relevant in the South Korean context, where misconceptions about HPV’s impact on males are prevalent, and there is limited engagement between healthcare providers and parents regarding male HPV vaccination [14]. Addressing these barriers through targeted public health campaigns and healthcare provider education is essential for ensuring the success of GNV [18].

## 4. Global Guidelines

In October 2023, the Advisory Committee on Immunization Practices (ACIP) introduced its updated immunization schedule for 2024, focusing on children and adolescents under 18 years of age in the United States [19]. The schedule, published by the Centers for Disease Control and Prevention (CDC), incorporates the latest recommendations to guide healthcare providers in following current vaccination practices. For HPV, the vaccine is typically recommended for boys and girls starting at 11 or 12 years old, with an option to begin at an age as early as 9. For those who missed the initial vaccination, a catch-up period extends until the age of 18 to ensure adequate protection. The HPV vaccine is administered as a two-dose series for those aged 9–14 years at initial vaccination, with doses being administered at 0 and 6–12 months, ensuring a minimum interval of 5 months between doses. For individuals > 15 years at the start of vaccination, a three-dose series is recommended, with doses being given at 0, 1–2 months, and 6 months, adhering to specific minimum intervals between doses. No additional doses are necessary once any HPV vaccine series, regardless of valence, has been completed according to the recommended dosing intervals [19].

The guidelines from the Advisory Committee on Immunization Practices (ACIP) align closely with those of the American Society of Clinical Oncology (ASCO) when it comes to cervical cancer prevention in high-resource environments. One key distinction, however, is ASCO’s recommendation for healthcare providers to begin offering the HPV vaccine at ages as early as 9 or 10. Additionally, ASCO advises against shared decision making for catch-up vaccination in adults over 27 years, pointing to minimal public health benefits for this age group and the difficulty in identifying individuals who would gain the most benefit from vaccination [20]. In contrast, recommendations for low-resource settings vary. The World Health Organization (WHO) focuses primarily on vaccinating girls aged 9 to 14, highlighting this group as the highest priority for HPV vaccination [18]. Vaccination for older women is only recommended if it remains affordable and cost-effective, ensuring it does not detract from efforts to vaccinate the primary target group or from essential cervical cancer screening programs (Table 2).

## 5. Benefits of Gender-Neutral Vaccination

The benefits of GNV in combating HPV-related diseases are profound and multifaceted. One of the primary advantages is enhanced herd immunity achieved through the inclusion of both men and women in vaccination programs. Studies have consistently shown that vaccinating both boys and girls not only directly protects those who are vaccinated, but also significantly reduces the prevalence of HPV within the broader population. Viral transmission is curtailed across both the male and female sexes, leading to fewer opportunities for the virus to spread and cause infection, thus decreasing prevalence.

Moreover, GNV has been demonstrated to provide incremental benefits beyond those achieved by vaccinating girls alone. A systematic review and meta-analysis of transmission-dynamic models [22] found that including men in HPV vaccination programs substantially increased the overall reduction in HPV prevalence among both men and women. For instance, with an 80% vaccination coverage, the relative reduction in HPV 16 prevalence was predicted to be 93% among women when both the male and female sexes are vaccinated compared to 83% with female-only vaccination [22]. This additional protection is crucial, particularly for MSM, a group that does not benefit from the herd immunity generated by female-only vaccination strategies [23]. Furthermore, GNV is associated with a greater potential for eliminating the HPV types targeted by the vaccine. The same meta-analysis revealed that with an 80% coverage in both boys and girls, the likelihood of eliminating HPV types 16, 18, 6, and 11 significantly increased [23]. This comprehensive approach to vaccination not only reduces the incidence of HPV-related cancers and diseases, but also facilitates the possible eradication of these high-risk HPV types in the future.

The benefits of GNV extend beyond the prevention of cervical cancer in women. By protecting boys against HPV, the program could significantly reduce the incidence of non-cervical cancers, including oropharyngeal and anal cancers [18]. However, as noted in the feedback, the current epidemiological models used to advocate for GNV in South Korea are based on outdated data and might not fully capture the evolving trends in HPV-related disease. Recent research has highlighted the significance of maintaining comprehensive disease and vaccine registries to monitor the long-term effectiveness of HPV vaccination in both males and females. In addition, while one-dose schedules have shown promise in middle-income countries like India, further research is needed to assess their applicability in South Korea [18].

In addition to herd immunity and direct protective benefits, real-world evidence reinforces the effectiveness of GNV in reducing the burden of HPV-related diseases. In Australia, the national implementation of GNV has led to a significant decline in the incidence of genital warts across both the male and female sexes, with an 89% reduction among young heterosexual men aged 15–20 years following the inclusion of boys [24]. This reduction, which surpasses the decrease observed during the female-only vaccination period, underscores the added value of vaccinating boys alongside girls. Moreover, the benefits of GNV extend beyond its immediate impact on genital warts.

Including males in HPV vaccination programs is essential because of their significant role in HPV transmission and their susceptibility to HPV-related diseases, such as oropharyngeal, anal, and penile cancers. Studies have shown that the HPV transmission rates from women to men are notably high, and men often develop lower natural immunity against HPV, making them more prone to persistent infections [25]. Vaccinating men not only protects them from these serious health risks, but also helps reduce the overall prevalence of HPV, thereby enhancing the effectiveness of public health efforts to control HPV-related diseases across the population.

The COVID-19 pandemic has disrupted HPV vaccination programs globally, raising concerns about the effectiveness of cervical cancer prevention. Recent modeling using the EpiMetHeos model [26] calibrated to Indian data showed that shifting from a female-only strategy to a GNV strategy can significantly enhance the resilience of these programs [27]. In a scenario with a 5-year disruption and 60% coverage before the disruption, GNV could prevent 302 cervical cancer cases per 100,000 girls born compared with only 107 cases in a female-only strategy, representing a 2.8-fold increase. At 90% coverage, the number of cases prevented increases by 2.2-fold from 209 to 464. Additionally, GNV meets the WHO’s cervical cancer elimination threshold by reducing the long-term incidence rate to 2.8 cases per 100,000 women per year at 60% coverage, compared to 4.7 cases in a female-only strategy. This highlights GNV’s critical role in enhancing the resilience and effectiveness of HPV vaccination.

## 6. Position Statement

To prevent HPV-related diseases, including cervical cancer, oropharyngeal cancer, and HPV-related benign disease, it is recommended to administer the HPV vaccine to males and females aged 9–26.Those between 9 and 14 years old should receive two doses at 0 and 6 months, while those between 15 and 26 should receive three doses at 0, 1, and 6 months.The recommended age for vaccination is 11–12 years old. Women aged 27 and older can receive the HPV vaccine at the discretion of healthcare providers.

## Figures and Tables

**Table 1 vaccines-12-01110-t001:** HPV vaccination coverage by birth year in South Korea (as of June 2023).

Birth Year	Female			Male		
	Eligible Population	Vaccinated with 1st Dose, %	Vaccinated with 2nd Dose, %	Eligible Population	Vaccinated with 1st Dose, %	Vaccinated with 2nd Dose, %
1953–1962	3,740,430	0.3	0.2	3,594,012	0.0	0.0
1963–1972	4,233,081	1.3	1.2	4,307,277	0.1	0.0
1973–1982	3,951,159	5.3	5.0	4,063,075	0.3	0.3
1983–1994	3,828,004	14.4	13.5	4,151,905	2.1	1.85
1995–2002	2,381,349	18.3	16.6	2,600,855	1.6	1.4
2003	236,652	64.7	56.3	254,854	0.6	0.5
2004	227,534	79.2	70.7	244,253	0.5	0.4
2005	209,828	89.2	79.8	224,873	0.5	0.4
2006	216,570	90.1	79.6	231,688	0.5	0.4
2007	240,461	90.9	82.4	254,224	0.5	0.5
2008	227,022	89.4	78.8	240,409	0.6	0.5
2009	216,994	85.4	68.2	230,396	0.6	0.4
2010	229,135	70.9	29.1	243,635	0.4	0.2
2011	230,891	20.2	0.2	243,292	0.2	0.1

**Table 2 vaccines-12-01110-t002:** HPV vaccination recommendations by Advisory Committee on Immunization Practices (ACIP) and World Health Organization (WHO).

**Advisory Committee on Immunization Practices (ACIP) [19]**
Routine HPV vaccination is recommended at 11–12 years of age. It can be administered starting at 9 years of age. For adolescents and adults aged 13–26 years who have not previously been vaccinated or completed the vaccine series, catch-up vaccination is recommended. For adults 27 years and older, catch-up vaccination is not routinely recommended; the decision to vaccinate people in this age group should be made on an individual basis.
**World Health Organization (WHO) [21]**
HPV vaccines should be included in national immunization programs, and the primary target is girls aged 9–14 years. Priority groups include immunocompromised individuals and those who have faced sexual abuse. Vaccination can be extended to secondary targets, such older women, boys, and men, only if feasible and affordable.

## Data Availability

The data are contained within the article.

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
