# Peer review of "Position Statement about Gender-Neutral HPV Vaccination in Korea"

_vaccines, 2024, doi:10.3390/vaccines12101110_

Round 1
Reviewer 1 Report
Comments and Suggestions for Authors
Response to Reviewers
High income countries, such as South Korea, are moving towards gender neutral HPV vaccine (GNV) programmes. South Korea has established a National Immunization Schools programme with high HPV coverage in girls and it is not surprising that calls to include boys are becoming more pressing. The current paper is clinically focused and provides the perspective of specialists who manage HPV-related disease in both sexes and understandably, see GNV as the way forward.
The authors argue a very general and uncritical case for GNV HPV programmes with little specific justification for such progression in South Korea. Apart from a passing reference to prevalence of head and neck cancers in this country in paragraph 2.4, it is not till paragraph 2.6 that this is somewhat briefly addressed. While some global context is certainly required, each country must consider its own situation, not just in terms of disease burden, but also the challenges of doubling the vaccine eligible population, and the potential acceptability of these changes to the general public. Reference to which the authors might have referred are:
Grandahl M and Neveus T. Barriers towards HPV vaccination for boys and young men: a narrative review. Viruses 2021, 13 (8) 1644.
Choi J, Cuccaro P Markham C et al. HPV vaccination intentions among Korean mothers of boys. Prev Med Reports Jan 2024, 37, 102566.
A few benefits of GNV are outlined in section 4 on Benefits, but this section is uncritical and over-stated: “Studies have consistently shown………” and the epidemiological models quoted are based on very early data. Reference 25 refers to a more recent model for India but it is not directly comparable as it is based on delivery of a one dose HPV vaccine schedule to both sexes in a middle income country where a high proportion of the population lives in poverty and has limited access to secondary care. Current research on implementing GNV approaches is lacking. A relevant factor for clinicians is that, while the National Programme has since 2002 built up an Immunization Registry Information System, the inter-relationship between vaccine and disease registers is not robust. Globally there is limited research on the effects of HPV vaccination on anal or oropharyngeal cancers.
Author Response
Thank you very much for your opinion. The relevant content is replaced with the attached file.

Reviewer 2 Report
Comments and Suggestions for Authors
The research addresses an important area in science and health.
Comments:
1. The manuscript is well-structured and provides a clear rationale for gender-neutral HPV vaccination in Korea. The flow from the introduction of HPV’s global burden to the focus on the Korean context is logical and well-supported with references. However, the transition between sections could be smoother, particularly from discussing global burdens to specific issues in Korea. Consider adding more explicit section headings to improve readability.
2. The manuscript relies heavily on peer-reviewed data and meta-analyses, which is commendable. However, to strengthen the evidence supporting the benefits of GNV, it would be beneficial to include more region-specific data from South Korea. This could involve providing information on vaccine coverage rates and outcomes from any pilot GNV programs if available. To enhance clarity, it would be helpful to include a table summarizing key statistics on HPV-related disease burdens in South Korea, especially for cancers such as oropharyngeal and anal cancers in men.
3. The manuscript addresses the increasing prevalence of oropharyngeal cancers in Korean men, emphasizing the importance of the HPV vaccination. However, the social and economic barriers to the implementation of GNV in Korea should be explored further. For example, are there any cultural or financial obstacles to expanding the vaccine program to boys? Discussing these issues would strengthen the argument for policy changes.
4. The statement might benefit from highlighting the current status of HPV vaccination in South Korea, including age groups and vaccination rates, to help understand the current landscape and the potential gaps GNV would address.
5. The exploration of HPV's impact on male fertility and sperm quality is an innovative and significant aspect. However, this section feels somewhat disjointed from the rest of the manuscript. Consider integrating these findings more directly into the broader argument for GNV. Perhaps creating a separate subsection or integrating the findings more cohesively with other sections discussing HPV's wider health impacts will be helpful.
6. Minor Suggestions: Some of the references are repeated, and citations such as [9], [11], [16], etc., are clustered together. It would be beneficial to spread out these references throughout the text, providing more specific citations for different parts of the argument.
7. Consider providing an executive summary or concluding section that clearly outlines the policy recommendations and next steps for implementing GNV in Korea.
Author Response
Thank you very much for your opinion. The relevant content is replaced with the attached file

Round 2
Reviewer 1 Report
Comments and Suggestions for Authors I have read through the revised manuscript and am satisfied with the revisions made by the authors. In my view, it could now be accepted for publication.